# Reservoir Effect of Textile Substrates on the Delivery of Essential Oils Microencapsulated by Complex Coacervation

**DOI:** 10.3390/polym16050670

**Published:** 2024-02-29

**Authors:** José Alexandre Borges Valle, Rita de Cássia Siqueira Curto Valle, Cristiane da Costa, Fabrício Bezerra Maestá, Manuel José Lis Arias

**Affiliations:** 1Department of Textile Engineering, Federal University of Santa Catarina, Florianópolis 88040-900, Brazilcristiane.costa@ufsc.br (C.d.C.); 2Textile Engineering (COENT), Universidade Tecnológica Federal do Paraná (UTFPR), Apucarana 86812-460, Brazil; fabriciom@utfpr.edu.br; 3INTEXTER-UPC, Universidad Politécnica de Cataluña, 08222 Terrassa, Spain; manuel-jose.lis@upc.edu

**Keywords:** essential oils, microcapsules, drug delivery, textiles

## Abstract

Microcapsules are being used in textile substrates increasingly more frequently, availing a wide spectrum of possibilities that are relevant to future research trends. Biofunctional Textiles is a new field that should be carefully studied, especially when dealing with microencapsulated essential oils. In the final step, when the active principle is delivered, there are some possibilities to quantify and simulate its doses on the skin or in the environment. At that stage, there is a phenomenon that can help to better control the delivery and the reservoir effect of the textile substrate. Depending on the chemical characteristics of the molecule to be delivered, as well as the structure and chemical nature of the fabric where it has been applied, there is physicochemical retention exerted by fibers that strongly controls the final rate of principle active delivery to the external part of the textile substrate. The study of this type of effect in two different substrates (cotton and polyester) will be described here regarding two different essential oils microencapsulated and applied to the substrates using padding technology. The experimental results of the final drug delivery demonstrate this reservoir effect in both essential oils.

## 1. Introduction

Scientific progress and technological development have significantly contributed to the improvement of people’s living standards. As a result, textiles have gained increased attention and more functions than clothing. Therefore, textiles can be used as drug carriers, which means they can be vehicles for the controlled release of pharmaceutical agents [1]. 

The fabrics used as a matrix (Figure 1) for the controlled release can be applied in many fields, from fashion to sportswear, cosmetics, protective clothing, and medical aid [2,3]. The potential techniques for incorporating active ingredients into fabrics are the application of microcapsules [4,5,6,7], cyclodextrins [8,9,10,11,12], liposomes [13,14,15], and metal–organic frameworks [16,17,18,19].

Microencapsulation is a versatile and flexible technical process used to provide a functional barrier to compounds (solids, liquids, gases, or a mixture of them) to avoid chemical and physical reactions and to maintain the biological, functional, and physicochemical properties of the core material [20]. It is also useful in improving the properties and usability of various industrial and commercial products [21]. 

There are many combinations between processes and polymers via which it is possible to obtain microcapsules, each of which imparts properties to the shell, such as stiffness, porosity, resistance, and chemical affinity with the medium and/or active substance [22]. Techniques can include simple or complex coacervation, fluidization, lyophilization, and spray-drying used to improve the stability of the active ingredients. Complex coacervation has been increasingly used in the methods used to form microparticles due to its simplicity, low cost, scalability, and reproducibility in encapsulation [23,24]. 

Regardless of the type of substance incorporated into the fabric, the release of the active compound can be accomplished via two methods: forced or controlled [25,26,27,28,29,30,31,32,33,34]. The second method is preferable as the controlled release system enables the release of these compounds under desired conditions, increasing the efficiency of the system [35,36].

However, it is important to emphasize that, for a textile substrate to have a pharmaceutical agent release function, the agent should have long-term stability, homogeneous surface charge, and a defined release rate in terms of time [37,38,39].

Thus, it is necessary to understand the controlled release mechanism as well as its complexity as the release of a substance may occur through more than one mechanism over time to determine matrix states [40].

In general, when the matrix structures come into contact with the dissolution media (or biological fluid), they can behave in two ways. First, they may maintain a constant structure throughout the dissolution process. Second, they may undergo swelling and, posteriorly, erosion [41,42,43].

There are still examples where these two matrix systems occur both by diffusion of the core and by biodegradation of the polymer. As in hydrogels, release is determined by both the rate of water absorption and the rate of diffusion through the swollen polymer matrix, the first phase being the rapid release and the second a slower phase [36,44]. In other cases, there may be a final phase, called the three-phase mechanism, where the final phase is attributed to the onset of polymer erosion [40,45].

The quantification of the kinetics and the determination of the release models, presented in the works of Jain et al. [46], can be performed in vitro using fixed parameters that are related to the media used in the system. In addition, it is possible to apply kinetic equations to verify the release mechanism. The most commonly used models are those of Higuchi [47] and Korsmeyer–Peppas [48].

Given the importance of the controlled release of microencapsulated active compounds, the study of the entire delivery phenomenon has been widely explored. The release mechanism depends on the type and amount of active compound, the type and amount of encapsulating material, the microcapsule preparation technique, the type and characteristics of the fabric, the environmental conditions during release, and the geometry and dimensions of the delivery system [49,50,51,52]. 

## 2. Drug Delivery Process 

The release from microcapsules’ tissue system can occur at a slow or first-order rate, or even with an initial burst release, followed by a slow or prolonged first-order release [49]. Many models, including empirical/semi-empirical and mechanistic realistic models, have been used to investigate the kinetics of drug delivery from controlled release formulations [49,50,51]. 

If the mechanism of transport of the active compound is purely diffusion-controlled, with constant diffusion coefficients, the mathematical treatment can be rather straightforward. Considering that the molecules of the active site are individualized or dissolved within the carrier material (what is called a monolithic solution) and in the absence of significant changes in the carrier matrix during drug release, the resulting release can be calculated depending on the system geometry. The release from microcapsules using the sphere approach is presented in Equation (1) [53]: (1)$$\frac{C_{t}}{C_{\infty }}=1-\frac{6}{\pi^{2}}\sum_{n=1}^{\infty }\frac{1}{n^{2}}exp\left(-\frac{Dn^{2}\pi^{2}t}{R^{2}}\right)$$
where *C_t_* and *C_∞_* denote the cumulative concentration of active compound released at time *t* and infinity, respectively; *D* is the diffusion coefficient of the active site within the matrix and *R* is the radius of the sphere.

When the release is considered from the plane geometry and semi-infinite media, it can be represented by Equation (2):(2)$$\frac{C_{t}}{C_{\infty }}=1-\frac{8}{\pi^{2}}\sum_{n=1}^{\infty }\frac{1}{{\left(2n+1\right)}^{2}}exp\left(-\frac{D{\left(2n+1\right)}^{2}\pi^{2}t}{L^{2}}\right)$$
where *L* is the thickness of the film. Equation (2) can be simplified for short times (Equation (3)) and long times (Equation (4)) to avoid the use of infinite series of exponential functions:(3)$$\frac{C_{t}}{C_{\infty }}=4\sqrt{\frac{Dt}{\pi L^{2}}},\;0\leq \frac{C_{t}}{C_{\infty }}\leq 0.6$$
(4)$$\frac{C_{t}}{C_{\infty }}=1-\frac{8}{\pi^{2}}exp\left(-\frac{D\pi^{2}t}{L^{2}}\right),\;0.4\leq \frac{C_{t}}{C_{\infty }}\leq 1.0$$

Equations (1)–(4) are applied for systems in which the release mechanism is controlled by diffusion, with no swelling and time-independent permeability for the active compound. If polymer swelling is of importance for the active release, which means that the length of the diffusion pathway increases with swelling or the mobility of the macromolecules significantly increases, resulting in increased mobility of the active compound, other models can be used to represent the release, such as those proposed by Korsmeyer and Peppas (Equation (5)) [54]:(5)$$\frac{C_{t}}{C_{\infty }}=kt^{n}$$
where *n* is the release exponent, an indication of the mechanism of release. A value of *n* equal to 0.5 results in the classical Higuchi equation and indicates diffusion-controlled release [55,56]. 

Setapa et al. 2020 [52] proposed a two-phase release model fitting for experimental data with considerably high swelling behavior. They obtained two diffusion coefficients, the first one being about 70–90% slower than the second; this condition was ascribed to the increase in the pore size of the devices as they swell, thereby resulting in easier and faster diffusion. However, if there are chemical interactions between the polymeric matrix of the textile substrate and the molecules of the active principle, these interactions will strongly influence the delivery rate, especially at medium time ranges, acting as a reservoir structure that modifies the mass transport of active principle molecules.

## 3. Experimental Procedure

### 3.1. Materials

Materials used for the formation of shell structure: chitosan of low molecular weight (50,000–190,000 Da, viscosity of 20–300 cP, degree of deacetylation ≥ 75%), and Arabic gum (Merck, Darmstadt, Germany). Tannic acid was used as a crosslinking agent (Merck, Darmstadt, Germany). The core active principles were lavender essential oil and citronella oil kindly gifted by Carinsa (Spain), and Tween 20 (Merck, Darmstadt, Germany). For the grafting reaction between the textile substrate and microcapsules, citric acid (Merck, Darmstadt, Germany) was used and monobasic sodium phosphate monohydrate (Panreac, Castellar del Vallés, Spain) as the catalyst. Textile substrates were woven fabrics, namely cotton fabric, and spun polyester type 54 fabric. 

### 3.2. Microcapsule Obtention

The microcapsule was produced by complex coacervation (Figure 2) between chitosan (1% *w*/*v*), Arabic gum (2% *w*/*v*), and oil in emulsion 2% (*v*/*v*). First of all, the biopolymer solutions were prepared separately. Chitosan was dissolved in acetic acid solution (0.1 M) with magnetic stirring for 15 min at 40 °C. Arabic gum solution was prepared in deionized water under the same stirring conditions at 40 °C to reach a complete dissolution. The lavender or citronella essential oil (2%) was stabilized in aqueous media with Tween 20 (2%) under stirring at 3000 rpm for 1 min (Ultraturrax T-25, IKA, Staufen, Germany). 

By mixing the chitosan and lavender oil solutions, the formation of the first polymer layer was configured and activation occurred to incorporate the second layer. After adjusting the pH to 3.5 with hydrochloric acid (0.2 M), the Arabic gum solution was added dropwise to the previous system. Once the mixture was achieved in magnetic agitation, it was emulsified for 1 min at a stirring rate of 8000 rpm (Ultraturrax T-25, IKA, Staufen, Germany). 

Finally, the temperature of the mixture was reduced to 5 °C, and the cross-linking agent, 2 mL of tannic acid (10% *v*/*v*), was added and kept at a constant temperature (5 °C) for 3 h. 

### 3.3. Impregnation of Textiles

The microcapsules obtained were applied onto cotton and polyester substrates by a conventional pad–dry–cure process using citric acid as a crosslinking agent [27]. A solution was prepared with each of the samples containing 10% (*w*/*v*) of the concentrated dissolution of microcapsules, 3% (*w*/*v*) citric acid, and 3% (*w*/*v*) monobasic sodium phosphate monohydrate. The textiles were immersed in the liquid mixture (1:20) for 10 min and, with a Foulard, the samples were impregnated at a pressure between rolls of 1 bar obtaining the wet pickup of 90%. Drying was carried out at 80 °C for 30 min. The samples were kept in a desiccator until the release assays.

## 4. Characterization of Microcapsules and the Microcapsule-Treated Textiles

Particle size distribution and zeta potential were measured by a laser particle size analyzer (Mastersizer-2000, Malvern, UK). To confirm that essential oils were successfully microencapsulated, an optical microscope (Olympus BX43F, Tokyo, Japan) was employed. The optical microscope was equipped with a digital camera controlled by analysis software and different objectives. 

The morphological analysis of the surface of the substrate was carried out using a scanning electronic microscope (Phenom Pro Desktop SEM Thermo Fischer, Waltham, MA, USA) without gold particle cover as particle impact did not change the surface characteristics of microcapsules. 

FTIR spectroscopy measurements were carried out on FTIR-8300 (Shimadzu, Kyoto, Japan) in the spectral region of 4000 to 500 cm^−1^ to verify the possible existence of interactions between the polymers conforming to the shell and the textiles. The compaction of microcapsules and possible interactions between the microcapsules and the textile substrate were studied using thermogravimetric analysis (TGA) thermal stability test (Mettler-Toledo, Columbus, OH, USA) performed at a heat rate of 10 °C min^−1^ in the temperature range of 30 °C to 800 °C in a nitrogen atmosphere with a flow of 50 mL min^−1^.

The yield on the release of essential oil in the microcapsules was performed by determining the quantity of the essential oil in the bath before and after the microencapsulation. The value was obtained by UV–Vis spectrophotometry (UV-2401, Shimadzu, Kyoto, Japan) at the maximum absorption wavelength and compared with a calibration curve by interpolation.

## 5. Release Experiments

The release behavior of citronella oil and lavender oil of the microcapsule-treated textiles was determined in triplicate following the methodology that was previously reported by Bezerra et al. [7] and Lis et al. [12] with some modifications. After the treatment with microcapsules, the textile substrates checked (2 cm × 2 cm) were placed into a vessel with 20 mL of deionized water and maintained in a thermostable bath at 37 °C ± 0.5 °C for approximately 300 min. Aliquots of 1.2 mL were extracted and mixed with absolute ethanol (1:24), and their absorbance was determined by spectroscopy at the ultraviolet range. In each essential oil, previous calibration curves were constructed using the same dilution factor. At each sample, the same volume was introduced, and the dilution was considered. 

## 6. Results and Discussion

### 6.1. SEM Measurements

The protocol used enables reaching a microencapsulation efficiency (initial concentration/final concentration/initial concentration) of 44.6% in the case of lavender oil and 57.3% regarding citronella oil. The final sizes of the microcapsules obtained are, on average, 15.6 nm and 25.6 nm, respectively. 

One can see the microcapsules of Citronella Oil on the surface of the fabric (Figure 3 (a) cotton and (b) polyester), and the microcapsules of Lavender Oil applied on the textile substrates (Figure 4 (a) Cotton and (b) PET). It can be seen that the morphology of the microcapsules on the surface is spherical and well-defined, and that they cover the fabric. The formation of small microcapsule agglomerates is due to the complex coacervation technique, as can also be seen in Bezerra’s work [6].

### 6.2. FTIR Measurements

The assessment of the microcapsules applied to the substrates by impregnation using FTIR is shown in the following figures:

The FTIR-ATR spectra of the lavender and citronella oil microcapsules are shown in Figure 5 (b and a, respectively). The bands at 3320 and 1621 cm^−1^ are characteristic of the presence of chitosan, and the bands between 661 and 900 cm^−1^ indicate the presence of Arabic gum. At the 1050 cm^−1^ range, the presence of lavender oil is detected. The citronella oil microcapsules, which can be seen in Figure 6a, show bands in the regions of 1636.74, 1545.55, and 1241.60 cm^−1^, which are characteristic of the primary, secondary, and tertiary amide groups [53,54,55,56,57,58,59], which confirm the formation of the coacervate since, during the coacervation process, the carboxylic groups of the polysaccharide interact with the amine groups of the chitosan [60]. 

Figure 6b shows the spectrogram in the attenuated infrared region of the cotton material after the treatment with microcapsules by the previously described Foulard method. It shows the presence of bands characteristic of cotton, such as 1001–1056 cm^−1^, 1205–1431 cm^−1^, and also the shift of peak 2922 (CH_2_ and CH_2_OH group vibrations) to 2900, which characterizes a change in the vicinity of these radicals and intermolecular interactions. 

However, the appearance of the 1540 cm^−1^ peak characteristic of secondary amide, a functional group present in microcapsules, after microcapsule treatment should be highlighted [6]. This makes it possible to verify the effectiveness of the finish.

Figure 6b shows the FTIR-ATR spectra of the cotton fabric treated with lavender oil microcapsules. As expected, the FTIR spectra show peaks characteristic of cellulose, with the stretching vibration of the hydroxyl group demonstrating a broad band at 3322 cm^−1^. The bands at 2894 cm^−1^ and 1158 cm^−1^ are characteristic of C–H stretching vibration and C–O–C stretching vibration, respectively. In addition, the small intensity of the peak at 1645 cm^−1^ can be attributed to the bending vibration of the NH group, resulting from the chemical interaction between the NH_2_ groups of the chitosan in the walls of the microcapsules and the COOH groups of the acids used as binding agents.

The analysis of the infrared spectrum of the treated 100% polyester fabric (Figure 6a) shows the presence of new bands in the 661 and 900 cm^−1^ regions, characteristic of the CH bond of aromatic groups present in the Arabic gum structure. There is also the presence of the 1450 cm^−1^ band, which refers to aromatic C=C alkenes present in gelatine, showing that there was only a surface treatment without any interaction.

The spectrum of the sample shows bands in the 720–870 cm^−1^ region, which refer to (CH_2_) _n_ angular deformation, vibrations of adjacent aromatic hydrogens, and vibration of the aromatic ring of the benzene functional group [61,62]. Other important bands present in the spectra are 1241 cm^−1^ of strong stretching of the C–O group of unsaturated and aromatic esters, 1338 cm^−1^ of angular deformation of the CH_2_ bending type, 1407–1471 stretching of the C-O group, and 1713 stretching of the C=O carbonyl group of the carboxylic acid groups. As can be seen, the surface modifications did not cause any changes when comparing the spectrograms.

### 6.3. Thermogravimetric Analysis

The samples have been submitted pertaining to the previously described protocol. Comparative results can be observed in the following figure (Figure 7).

Just to summarize the most important parameters obtained in these thermograms, Table 1 presents the evolution of T_onset_ (the first change on the slope) and T_offset_ (the last change on slope) and the mass loss (in percentage) at the T_max_.

The TGA results are shown in Table 1 and Figure 7. Basically, there is a single degradation stage for cotton, and, for polyester, there is a very pronounced loss of mass, close to 290 °C and 370 °C for cotton and polyester, respectively. 

In the case of cotton, this loss of mass corresponds to the depolymerization of cellulose chains. In the degradation stage, there was a greater loss of mass for the PES fabrics.

The final residue was similar for cotton and cotton with microcapsules; for PES, the microcapsule generates a protective effect, with more residue remaining at 540 °C. The presence of the microcapsule in the fabrics also did not alter the T_onset_, and neither did it alter the rest of the parameters in Table 2, which shows that the microcapsule did not alter the thermal stability of the material.

In both cases, cotton and PES, the effect of the presence of microcapsules does not modify the main parameters of the thermograms, keeping the principal thermal behavior correspondent to the substrate. The exception to this fact is the increase in char at the end of the thermogram. PES retains the majority of microcapsules with organic shells, which promotes unusual accumulation at the surface. On the contrary, the capability of absorption of bath by COT dilutes the effect of microcapsules that are better distributed inside the textile substrate.

### 6.4. Drug Delivery

From the experimental procedure described previously, the delivery of the active principle in the bath is shown in the following figures (Figure 8 and Figure 9).

There are two different regions of delivery behavior. In each case, the first step is controlled by diffusion regarding the linear tendency at short times. Until values of 16 min, PES acts by supplying the media in the control of the plain fabric. In the case of COT, differences in chemical characteristics show linearity until 4 min. Between the 4 and 16 min. range, COT fibers show a second linear step, which sometimes is confused with the main tendency and is related to the interactions of the essential oil with polymeric chains. This type of interaction between fabric and essential oil changes the mechanisms of global delivery from the control of diffused components from spherical microcapsules towards the main control exerted by the global exposed surface of the tissue. The delay produced on the tissue by the chemical affinity of active molecules with the polymeric chains of the substrate is called the reservoir effect.

To quantify the delivery results, the Korsmeyer–Peppas [54] model and the Higuchi model were fitted to the release data of lavender oil (Figure 9) and the delivery behavior of citronella oil (Figure 10). The parameters for both models were estimated using the LSQCURVEFIT function in MATLAB (7.0, MathWorks, Natik, MA, USA), and the routine Global Search was used to ensure that the minimum was not local (Table 1).

With the application of the equations mentioned before, the results can be observed in the following Table 2.

Good fitting of the experimental data was obtained for both the Higuchi and Korsmeyer–Peppas models [54]. The classical Higuchi model was better for cotton fabric in the first step of lavender oil release. For polyester, the Korsmeyer–Peppas model [54] resulted in a release exponent (*n* value) higher than 0.5, indicating the occurrence of simultaneous phenomena for release, such as diffusion and swelling. In slab geometry, when the value of *n* is between 0.5 and 1, it is called anomalous or non-Fickian transfer. In this case, the mechanism of diffusion is both swelling and diffusion. Polymeric chains rearrange slowly, and, at the same time, the diffusion process results in a time-dependent anomalous effect. Also, if polymer swelling is the sole release rate controlling mechanism (system with film geometry), zero-order release kinetics are observed, corresponding to a release exponent of *n* = 1 [50,57]. So, as the n value approaches 1, it is expected that swelling is more important for the release system.

In polymeric matrices in which the glass transition temperature is lower than the release medium temperature, polymeric chains move easily so that solvent penetration is enhanced and diffusion is facilitated. The same can be observed for polymeric matrices with T_g_ higher than the medium temperature when a plasticizer is added [58]. The T_g_ of dry cotton fabric is about 220 °C, with this value being decreased to below zero when saturated with water since it acts as a plasticizer for cotton [59]. So, it is expected that diffusion is more predominant in the release from cotton fabrics when saturated with water.

In the case of citronella oil (Figure 10), differences can be observed only in the diffusion-controlled stage where PES presents a higher rate of delivery. As was described in the lavender oil case, the values of the square root of time where linearity can be observed are, as well, short times. Until 16 min, there is a first step that extends with different slope until approximately 60.8 min.

Meanwhile, the COT substrate shows the first delay until 6.25 min, a clear first slope until 25 min, and another one until 49 min. From 100 min, basically, both fibers show the same behavior. This is different than for lavender oil, where differences observed from short times can be extended to long times.
(6)$$\frac{M_{\left(i-l\right)}}{M_{\infty }}=\;{k\left(t-l\right)}^{n}$$

The Korsmeyer–Peppas model [54] and the Higuchi model were also fitted to the release data of citronella oil (Figure 10 and Table 3). 

There are three main assumptions considered in the Korsmeyer–Peppas model: first, the general model is best suited for short times where C/C_∞_ < 0.6 in the release curve, which should be used for the determination of the exponent *n*; the ratio between the system length and thickness should be at least equal to 10 [54]. For the studied release systems, different maximum values for C/C_∞_ were considered in the model fitting (Figure 11). So, it was shown that lavender oil presented good fitting until values of 0.64 and 0.90 when using cotton and polyester fabrics, respectively. For citronella oil, the model was best suited for shorter times considering the release curves for C/C_∞_ < 0.4.

## 7. Conclusions

After these experimental results were obtained, it became clear that the influence of the textile substrate used is of crucial importance on the final behavior of the system on delivery of the essential oil introduced inside the fibers through microcapsules. The influence of the substrate is related, not only to the mutual chemical character of the shell structure of the microcapsule but also to the chemical characteristics of the components of the essential oil, versus the textile substrate. From Table 4, it is clear that, although the interaction between the external structures of the microcapsules were only interacted in cotton, the effect of polyester in the global transport coefficient is different attending the essential oil applied. Moreover, the effect of more uniform absorption of cotton substrates detected in TGA results (see Figure 6) is diluted when lavender oil is changed by citronella oil, using the same microcapsule structure.

The delay detected can be attributed to an extra effect that we have named as “reservoir effect”, due to the chemical mutual affinity shown between essential oil and substrate, apart from the diffusion-controlled first step.

## Figures and Tables

**Figure 1 polymers-16-00670-f001:**
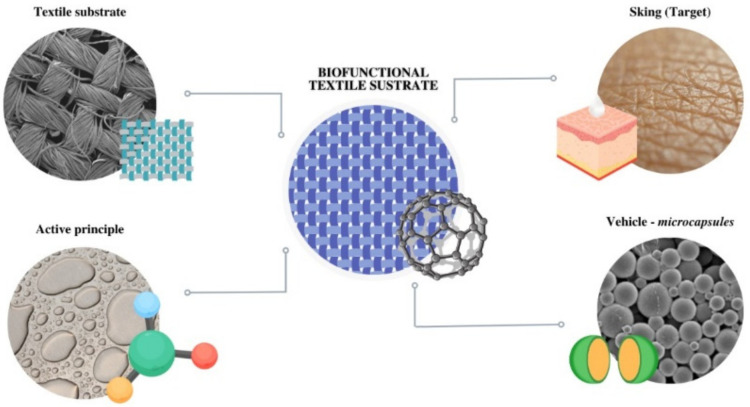
Global scheme of the application of microcapsules in textile substrates as vehiculizers of active principles to the skin.

**Figure 2 polymers-16-00670-f002:**
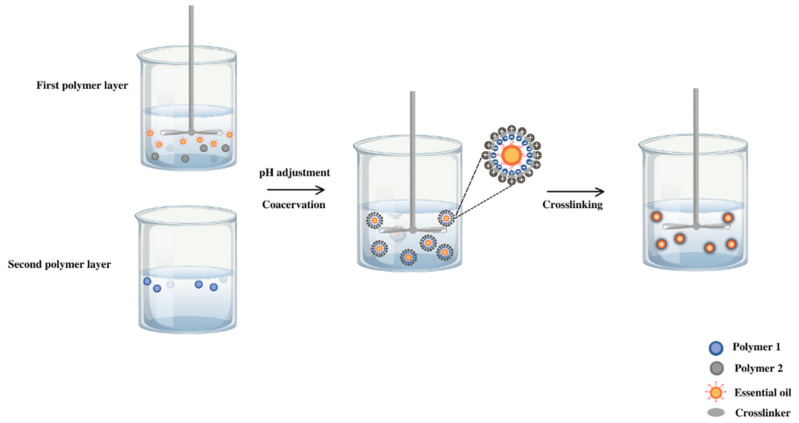
Experimental procedure for the coacervation method applied to citronella oil and lavender oil.

**Figure 3 polymers-16-00670-f003:**
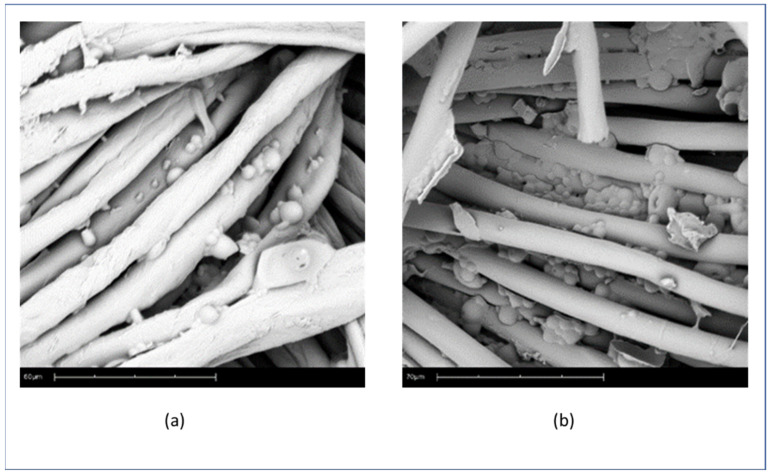
SEM pictures of microcapsules of citronella oil on the surface of fabric: (**a**) cotton (60 micra scale) and (**b**) polyester (70 micra scale).

**Figure 4 polymers-16-00670-f004:**
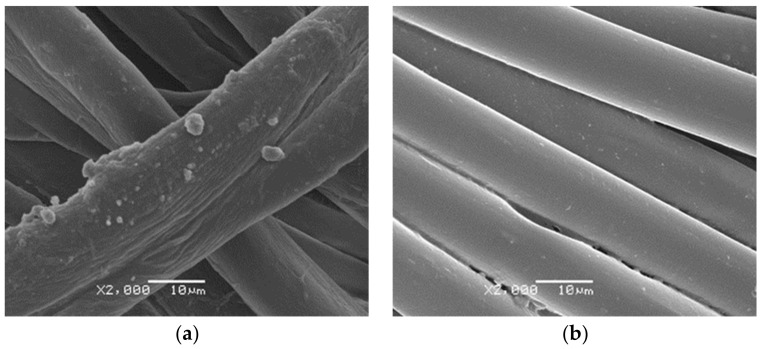
SEM pictures of lavender oil microcapsules on textile substrates (**a**) Cotton and (**b**) PET.

**Figure 5 polymers-16-00670-f005:**
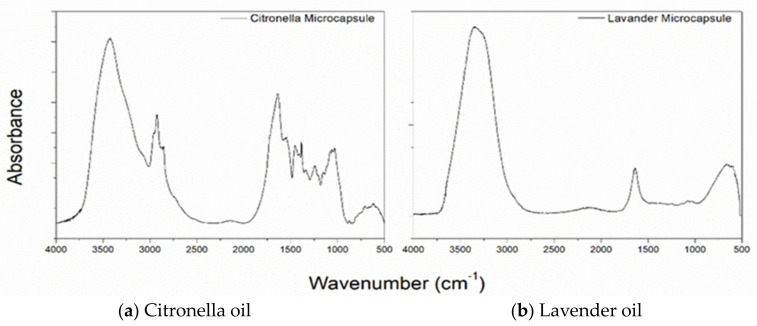
FTIR Spectra of microcapsules of (**a**) citronella oil and (**b**) lavender oil.

**Figure 6 polymers-16-00670-f006:**
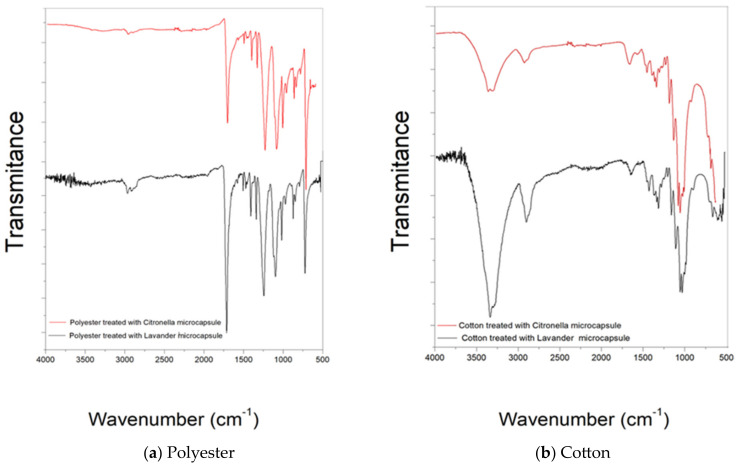
FTIR spectra of microcapsules applied to textile substrates by impregnation and Foulard: (**a**) polyester and (**b**) cotton.

**Figure 7 polymers-16-00670-f007:**
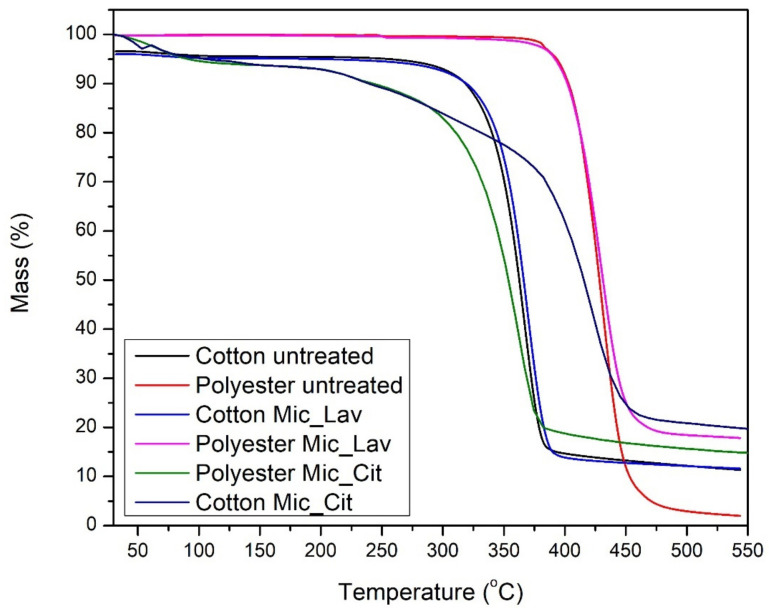
Thermograms of samples: cotton and polyester non-treated (black and red lines); cotton substrate with lavender oil microcapsules (brilliant blue) and citronella oil microcapsules (dark blue); polyester with microcapsules of lavender oil (rose) and citronella oil microcapsules (green).

**Figure 8 polymers-16-00670-f008:**
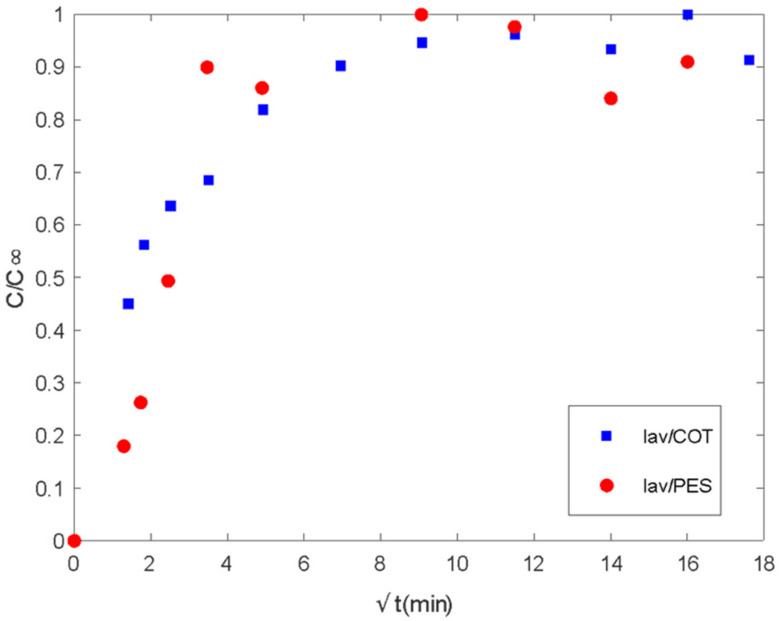
Delivery of lavender oil microencapsulated with chitosan–Arabic gum from PES and COT fabrics. *Ct/Cinf* vs. the square root of time (min^1/2^).

**Figure 9 polymers-16-00670-f009:**
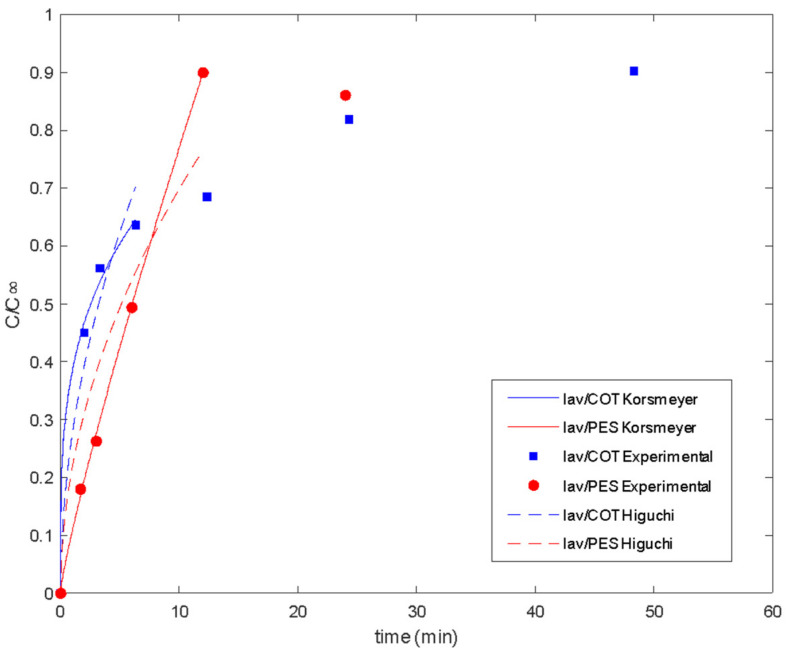
Delivery of lavender oil microencapsulated with chitosan–Arabic gum from PES and COT fabrics: experimental data and fitted models.

**Figure 10 polymers-16-00670-f010:**
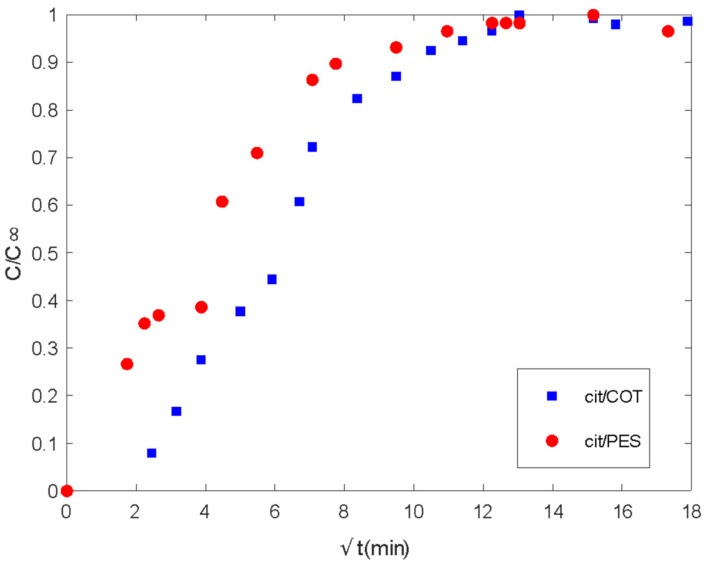
Delivery of citronella oil microencapsulated with chitosan-Arabic gum from PES and COT fabrics. *Ct/Cinf* vs. the square root of time (min).

**Figure 11 polymers-16-00670-f011:**
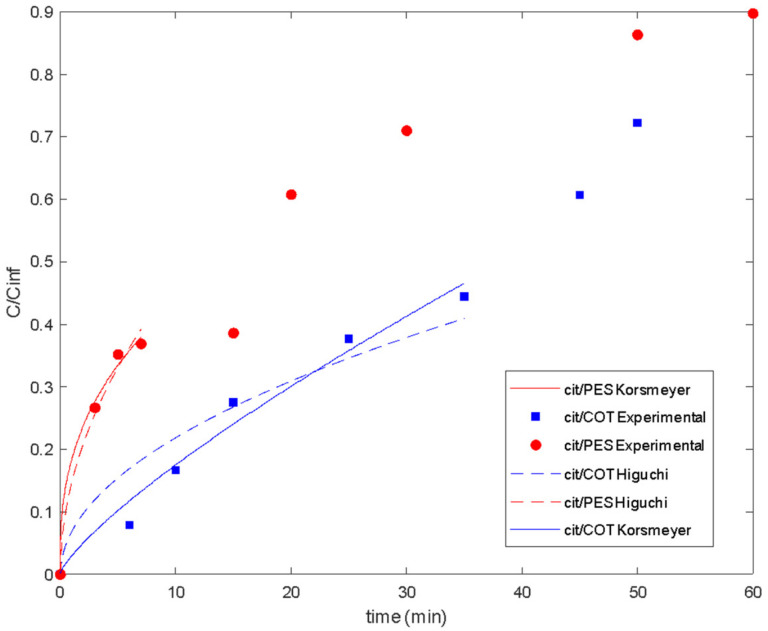
Delivery of citronella oil microencapsulated with chitosan–Arabic gum from PES and COT fabrics: experimental data and fitted models.

**Table 1 polymers-16-00670-t001:** Thermogram parameters of each textile sample.

Sample	T_onset_ (°C)	T_offset_ (°C)	T_max_ (°C)	Mass Loss (%)	Residue at 540 °C
Cotton Untreated	290	390	362	78.557	11.40
Cotton micro_Lav	290	407.7	373.8	79.663	11.60
Cotton micro_Cit	288	396	363	80.25	11.10
PET Untreated	373	490	425	95.604	2.00
PET micro_Lav	357	490	425	80.127	17.80
PET micro_Cit	353	480	425	82.88	15.00

**Table 2 polymers-16-00670-t002:** Estimated parameters for lavender oil release using the Korsmeyer–Peppas (K–P) [54] model and the Higuchi model.

	Model	K	n	Sq.Res.norm.	R^2^
Lav/COT	K–P	0.3829 ± 0.1121	0.2825 ± 0.2018	8.8629 × 10^−4^	0.9973
	Higuchi	0.2798 ± 0.0544	-	0.0102	0.9686
Lav/PES	K–P	0.1066 ± 0.0153	0.8577 ± 0.0638	3.3145 × 10^−4^	0.9994
	Higuchi	0.2212 ± 0.0624	-	0.0459	0.9237

**Table 3 polymers-16-00670-t003:** Estimated parameters for citronella oil release using the Korsmeyer–Peppas model and the Higuchi model.

	Model	K	n	Sq.Rs.nor.	R^2^
C_it_/COT	K–P	0.0288 ± 0.0254	0.7827 ± 0.2724	0.0036	0.9800
	Higuchi	0.0693 ± 0.0139	-	0.0132	0.9265
C_it_/PES	K–P	0.1824 ± 0.1099	0.3755 ± 0.3606	5.0271 × 10^−4^	0.9957
	Higuchi	0.1484 ± 0.0154	-	0.0011	0.9909

**Table 4 polymers-16-00670-t004:** Global mass transport coefficients estimated using Equation (3).

	D/L^2^ (min^−1^)
Lav/COT	0.0154 ± 0.0060
Lav/PES	0.0096 ± 0.0054
Cit/COT	0.0009 ± 0.0003
Cit/PES	0.0043 ± 0.0009

## Data Availability

Data can be supplied by petition.

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
