# Peer review of "Reservoir Effect of Textile Substrates on the Delivery of Essential Oils Microencapsulated by Complex Coacervation"

_polymers, 2024, doi:10.3390/polym16050670_

Round 1

Reviewer 1 Report

Comments and Suggestions for Authors

This article, the authors talked about the use of microcapsules in textile substrate. They fabricated different composition microcapsules for design of textile substrate, such as using the chitosan as microcapsule materials and lavender citronella oil. Before publication, Here are some issues need to be addressed.

1.     In this article, the authors talked about the microcapsules, but there are no optical images or SEM images of microcapsules. The images which prove the structure of the microcapsules are needed. And where’s the images of the microcapsule-treated textiles such as cotton and polyester.

2.     The quality of Figure 4 is poor, which needs to be improved. And some images about the fabricating process of the microcapsules and microcapsule-treated textiles are needed.

3.     What’s the diameter of the microcapsules? And what’s the encapsulation efficiency of the microcapsules?

4.     In the Figure 1, the authors showed blue lipid vesicles, but in this article, the authors didn’t use it, please correct the image to the real microcapsule model, and the figure captions are needed.

Comments on the Quality of English Language

Good.

Author Response

Dear reviewer

Many thanks for your advices in the revision process. We have tried to follow your comments to improve the article.

Reviewer 2 Report

Comments and Suggestions for Authors

In this study, the authors have evaluated the reservoir effect of textile substrates in the delivery of essential oils microencapsulated by complex coacervation. However, one of the major flaws is that the experimental design and results are rather simple, making the manuscript lack strong elucidation to support its novelty and claims. The literature review is long-winded but not strong enough to provide research gaps. In addition, the English language used in the manuscript needs major improvements. It contains numerous grammatical errors that even do not allow the reader to focus completely on the substantive content of the article. As a result, I don't recommend this manuscript to be accepted in its current state. 

Some specific comments:

1. The abstract should state briefly the purpose of the research, the principal results and major conclusions. The current Abstract looks like an introduction and should be rewritten.

2. The presented introduction is too interminable to understand the significance and objectives of this work. I strongly recommend the authors polish or rewrite this section.

3. As one of the most important parts of this manuscript, complex coacervation has been neglected by the authors! You should add a brief intro about complex coacervation, its importance in your work, unique properties that convinced you to use it in your research, and how complex coacervation contributes to the novelty of this work?

4. 2. Drug-delivery process to skin: It is unclear why you added this section here?

5. “Chitosan of low molecular weight and Arabic Gum” How to define the low molecular weight? And what is the degree of deacetylation of the Chitosan? Please provide sufficient technical details to allow others to reproduce the work.

6. “Drying was carried out at 80 â—¦C for 3 min.”  Do you think 3 min is enough for drying the samples?

7. “After the treatment with microcapsules, the textile substrates checked (2 cm x 2 cm) were placed into a vessel with 20 mL of deionized water” Please specify how to treat with microcapsules? Also, here you should use PBS rather than deionized water. This experiment has to be redone.

8. Figure 6: FT-IR Spectra of cotton treated with different microcapsules should be merged to clear the differences.

Comments on the Quality of English Language

English language used in the manuscript needs major improvements. It contains numerous grammatical errors that even do not allow the reader to focus completely on the substantive content of the article.

Author Response

Dear reviewer

Many thanks for your extense revision and your useful advices. We have followed all of your indications to improve the article

Round 2

Reviewer 1 Report

Comments and Suggestions for Authors

The authors have sufficiently improved their paper, I suggest the manuscript to publish now.

Author Response

Many thanks for your kind words

Reviewer 2 Report

Comments and Suggestions for Authors

The revised manuscript has been improved but there are still some issues that have not been addressed yet. Here are some remarks on the revised manuscript:

1. Per my previous comment, the FT-IR Spectra of cotton treated with different microcapsules should be merged to clear the differences. However, the authors replied that “FT-IR spectra have been obtained in different formats in different places of the world.” This reply is unacceptable. I still suggest that these FT-IR Spectra should be merged. In addition, the quality of FT-IR Spectra is poor.

2. What is the scale bar for current Figure 3?

3. Line 356: “From the experimental procedure, described previously, the delivery of the active principle in the bath is shown in the following Figures (Fig 7, Fig. 9)”. Here it should be Fig.8 and Fig. 9, not Fig 7.

Comments on the Quality of English Language

Extensive editing of English language required.

Author Response

Many thanks for your advice. We have proceeded to change following your advices
